# Comparative Study of Variations in Quantum Approximate Optimization Algorithms for the Traveling Salesman Problem

**DOI:** 10.3390/e25081238

**Published:** 2023-08-21

**Authors:** Wenyang Qian, Robert A. M. Basili, Mary Mehrnoosh Eshaghian-Wilner, Ashfaq Khokhar, Glenn Luecke, James P. Vary

**Affiliations:** 1Instituto Galego de Fisica de Altas Enerxias (IGFAE), Universidade de Santiago de Compostela, E-15782 Santiago de Compostela, Spain; 2Department of Physics and Astronomy, Iowa State University, Ames, IA 50011, USA; jvary@iastate.edu; 3Department of Electrical and Computer Engineering, Iowa State University, Ames, IA 50011, USA; basiliro@iastate.edu (R.A.M.B.); ashfaq@iastate.edu (A.K.); 4Department of Mathematics, Iowa State University, Ames, IA 50011, USA; mmew@iastate.edu (M.M.E.-W.); grl@iastate.edu (G.L.)

**Keywords:** quantum computing, quantum simulation, quantum approximate optimization algorithm, traveling salesman problem, noisy simulation

## Abstract

The traveling salesman problem (TSP) is one of the most often-used NP-hard problems in computer science to study the effectiveness of computing models and hardware platforms. In this regard, it is also heavily used as a vehicle to study the feasibility of the quantum computing paradigm for this class of problems. In this paper, we tackle the TSP using the quantum approximate optimization algorithm (QAOA) approach by formulating it as an optimization problem. By adopting an improved qubit encoding strategy and a layer-wise learning optimization protocol, we present numerical results obtained from the gate-based digital quantum simulator, specifically targeting TSP instances with 3, 4, and 5 cities. We focus on the evaluations of three distinctive QAOA mixer designs, considering their performances in terms of numerical accuracy and optimization cost. Notably, we find that a well-balanced QAOA mixer design exhibits more promising potential for gate-based simulators and realistic quantum devices in the long run, an observation further supported by our noise model simulations. Furthermore, we investigate the sensitivity of the simulations to the TSP graph. Overall, our simulation results show that the digital quantum simulation of problem-inspired ansatz is a successful candidate for finding optimal TSP solutions.

## 1. Introduction

For over a century, the traveling salesman problem (TSP) [1] has inspired hundreds of works and dozens of algorithms, of both exact and heuristic approaches. Today, the TSP has become so quintessential in modern computing that it is commonly considered the prototypical NP-hard combinatorial optimization problem, possessing far-reaching impact on countless applications in science, industry, and society. Consequently, the TSP is frequently taken as an ideal candidate for new computational models and non-standard algorithmic approaches, including approximate approaches like simulated annealing [2] and self-organizing maps [3], which have been widely employed to tackle the TSP.

Recent advancements in quantum technologies have paved the way for various quantum computing approaches to tackle the traveling salesman problem (TSP). These approaches include the quantum Held–Karp algorithm [4], quantum annealing (QA) [5,6,7,8,9], and the more general variational quantum algorithm [10,11] (VQA). VQA approaches have found extensive applications in diverse fields such as chemistry [11], physics [12], and finance [13], among others. Although complete demonstrations of quantum advantage over classical algorithms are currently limited due to the noisy intermediate-scale quantum (NISQ) era [14], exploring these quantum algorithms remains crucial, as experimentation on prototype quantum hardware continues to rapidly approach what can be classically simulated by even the world’s largest supercomputers. Notably, the quantum approximate optimization algorithm (QAOA) [10,15], a subclass of the general VQA, has been successfully applied to a number of optimization problems [16], including the max-cut problem [17,18], vehicle routing [19], DNA sequencing [20], protein folding [21], as well as the TSP [22]. In comparison to the popular hardware-efficient VQA, the QAOA takes advantage of the domain knowledge of the specific problem at hand to produce a variational ansatz with fewer parameters and a shallower depth. Furthermore, an extension of the original QAOA called the quantum alternating operator ansatz [23,24] offers a generalized approach that specializes in solving problems with hard constraints.

In the NISQ era, the QAOA approach can be particularly advantageous for addressing the challenges of the traveling salesman problem (TSP), owing to the QAOA’s hybrid feature, hardware-friendly structure, and controlled optimization. Being a hybrid approach, the QAOA exhibits robust tolerance to systematic errors by leveraging classical computer optimizers. Its layered ansatz structure inspired by the problem Hamiltonian allows for high flexibility in the circuit depth and qubit coherence time, incorporating the capabilities offered by the quantum backends. Compared with the QA [25,26], the QAOA also enables fine control of optimization through its finite layers, which is particularly beneficial in the current NISQ era. However, the numerical simulation of the QAOA on the TSP, especially in the multiple-layer region, is not well understood, since the non-adiabatic mechanism of the QAOA differs significantly from that of the QA [27]. Therefore, it becomes imperative to explore various implementations of the QAOA to determine the optimal path for simulation. Conducting investigations into these problems on digital quantum computers or simulators is essential, as they have the potential to unveil new quantum simulation strategies for traditional optimization tasks. We distinguish the present work from the previous studies by constructing our QAOA using different ansatzes and comparing their performances in both numerical accuracy and resource cost, which addresses a crucial aspect that is often neglected in conventional studies.

In this work, we study the effectiveness of three distinct designs of the QAOA in solving the TSP by adopting a layer-wise learning optimization protocol [28] on digital quantum simulators via Qiskit [29]. We organize this paper as follows: In Section 2, we introduce the TSP and its mathematical formulation as a binary constraint optimization problem. In Section 3, we outline the QAOA methods, with particular focus on the initialization, mixer ansatz, and measurement protocol employed in this work. In Section 4, we present and compare the numerical results of the QAOA simulation on TSP instances with 3, 4, and 5 cities, utilizing different ansatz designs. We discuss the impact of the device noise and TSP variations on the simulation results. In Section 5, we summarize the results and discuss plans for the future.

## 2. Traveling Salesman Problem

In this section, we first define the TSP as an optimization problem and then improve its formulation by taking advantage of the symmetry in the solution.

### 2.1. TSP Formulation as an Optimization Problem

The traveling salesman problem asks for the shortest path that visits each city exactly once and returns to the starting city. In the symmetric case where the distance between any two cities is the same regardless of the traveling direction, the TSP can be reformulated as an undirected graph problem where its vertices represent cities and the edge weights represent traveling distances. Mathematically, given an undirected graph *G* with vertices *V* and edges *E*, i.e., G=(V,E), we aim to find a Hamiltonian cycle that goes through all |V| nodes exactly once with the smallest total weights of the connecting edges on the path.

In this graph formulation of the TSP, any valid cycle, be it minimum or not, can be represented by a visiting order or a permutation of integers, such as {0,1,…,n−1}, where the integers are the city indices starting at 0 for a total of *n* cities. Alternatively, the visiting order on a TSP graph can be conveniently described by a sequence of binary decision variables $x_{i,t}$, indicating whether city-*i* is visited at time *t* [30]. If $x_{i,t}=1$, then city-*i* is visited at *t*; otherwise, the city is not visited by the traveling salesman. Naively, to fully describe the solution to an *n*-city TSP, a total of $n^{2}$ binary variables is needed in this representation.

Alternatively, this “one-hot” representation of binary decision variables can be written collectively in either a matrix or flattened array format for numerical implementation. For instance, a valid Hamiltonian cycle of permutation x=(0,1,2,3) is translated into binary decision variables *x* as:(1)$$\begin{array}{c} x=\left(0,1,2,3\right)\equiv \left(\begin{array}{cccc} 1 & 0 & 0 & 0\\ 0 & 1 & 0 & 0\\ 0 & 0 & 1 & 0\\ 0 & 0 & 0 & 1 \end{array}\right)\equiv 1000010000100001, \end{array}$$
where the matrix row index represents each city index, and the column index represents each time instance. City-*i* is visited at time *t* if and only if $x_{i,t}=1$. In this work, all three notations (permutation, matrix, and bit string array) are used interchangeably. Any Hamiltonian cycle in the TSP has a unique sequence of binary decision variables or “bit string". However, the reverse is not true, since a large portion of the possible bit strings may not correspond to any meaningful permutation. Specifically, we classify any bit string *x* into three categories or states:(2)$$\begin{array}{cc} x & =\left\{\begin{array}{cc} \mathrm{true}, & x\;\mathrm{is}\;a\;\mathrm{permutation}\;\mathrm{and}\;\mathrm{gives}\;\mathrm{the}\;\mathrm{shortest}\;\mathrm{path},\\ \mathrm{false}, & x\;\mathrm{is}\;a\;\mathrm{permutation}\;\mathrm{but}\;\mathrm{does}\;\mathrm{not}\;\mathrm{give}\;\mathrm{the}\;\mathrm{shortest}\;\mathrm{path},\\ \mathrm{invalid}, & x\;\mathrm{is}\;\mathrm{not}\;a\;\mathrm{permutation}, \end{array}\right. \end{array}$$
where the true and false bit strings are also called valid bit strings. Any bit string can be translated to a Hamiltonian cycle if and only if it is a permutation. Clearly, invalid solutions are disallowed traveling orders to the TSP.

With binary decision variables *x*, a true solution to an *n*-city TSP can be found by finding an *x* that minimizes the following cost function [30]:(3)$$\begin{array}{c} C_{\mathrm{dist}}\left(x\right)=\sum_{0\leq i,j<n}\omega_{ij}\sum_{t=0}^{n-1}x_{i,t}x_{j,t+1}, \end{array}$$
where $\omega_{ij}$ is the distance (or edge weight in the undirected graph) between city-*i* and city-*j* (in the symmetric TSP, $\omega_{ij}=\omega_{ji}$ and $\omega_{ii}=0$). Here, $C_{\mathrm{dist}}\left(x\right)$ also gives the shortest TSP distance when *x* is a true solution. Since the cost function itself does not forbid invalid solutions in general, additional constraint conditions must be satisfied for a valid Hamiltonian cycle, such as: (4)$$\begin{array}{cc} \sum_{i=0}^{n-1}x_{i,t} & =1\;\mathrm{for}\;t=0,1,\ldots ,n-1 \end{array}$$(5)$$\begin{array}{cc} \sum_{t=0}^{n-1}x_{i,t} & =1\;\mathrm{for}\;i=0,1,\ldots ,n-1, \end{array}$$
where Equation (Equation 4) forbids multiple cities visited by the traveler at the same time, and Equation (Equation 5) forbids revisiting the same city. Alternatively, in the matrix format, these constraints are easily implemented by requiring that any row or column sum to exactly one. These two hard constraints are the necessary conditions for any valid solution, though not necessarily a true solution to a TSP. To formulate the TSP as a minimum-optimization problem, these constraint conditions are conveniently incorporated as the penalty terms, such that the combined cost function, C(x) becomes:(6)$$\begin{array}{cc} C(x)= & \;C_{\mathrm{dist}}\left(x\right)+\lambda C_{\mathrm{penalty}}\left(x\right) \end{array}$$(7)$$\begin{array}{cc} = & \sum_{0\leq i,j<n}\omega_{ij}\sum_{t=0}^{n-1}x_{i,t}x_{j,t+1}+\lambda \left\{\sum_{t=0}^{n-1}{\left(1-\sum_{i=0}^{n-1}x_{i,t}\right)}^{2}+\sum_{i=0}^{n-1}{\left(1-\sum_{t=0}^{n-1}x_{i,t}\right)}^{2}\right\}, \end{array}$$
where λ is the weight factor of the penalty term, serving as the Lagrange multiplier, which should be positive and sufficiently large. It is easy to see that bit string *x* gives the minimum of C(x) if and only if *x* is a true solution to the given TSP. Finding a Hamiltonian cycle to the TSP is now equivalent to finding an $x^{*}$ that minimizes C(x) in Equation (Equation 6), i.e., $x^{*}=\arg\min\;C\left(x\right)$.

### 2.2. Improved TSP by Eliminating Rotational Symmetry

Symmetry plays a vital role in many graph optimization problems, and exploiting them can help reduce the problem’s complexity. In the previously introduced TSP optimization, one uses $n^{2}$ decision variables for *n* cities. However, solutions obtained after the optimization display “rotational” symmetry; they are physically identical up to some rotation. For example, a visiting order of permutation (0,1,2) is equivalent to (1,2,0) and (2,0,1) for a three-city TSP. They form a natural equivalence class for the solution sets. To reduce the size of the search space (and the number of qubits to encode), a simple but significant improvement can be made by fixing the starting city [30].

Without loss of generality, we fix city-0 as our starting point. The traveling salesman will return to city-0 after visiting all other cities exactly once. Then, the improved cost functions $C_{\mathrm{dist}}^{\prime }\left(x\right)$ and $C^{\prime }\left(x\right)$ become: (8)$$\begin{array}{cc} C_{\mathrm{dist}}^{\prime }\left(x\right)= & \sum_{1\leq i,j<n}\omega_{ij}\sum_{t=1}^{n-2}x_{i,t}x_{j,t+1}+\sum_{i=1}^{n-1}\omega_{0i}\left(x_{i,1}+x_{i,n-1}\right), \end{array}$$(9)$$\begin{array}{cc} C^{\prime }\left(x\right)= & \;C_{\mathrm{dist}}^{\prime }\left(x\right)+\lambda C_{\mathrm{penalty}}^{\prime }\left(x\right) \end{array}$$(10)$$\begin{array}{c} \begin{array}{cc} = & \sum_{1\leq i,j<n}\omega_{ij}\sum_{t=1}^{n-2}x_{i,t}x_{j,t+1}+\sum_{i=1}^{n-1}\omega_{0i}\left(x_{i,1}+x_{i,n-1}\right)\\ + & \lambda \left\{\sum_{t=1}^{n-1}{\left(1-\sum_{i=1}^{n-1}x_{i,t}\right)}^{2}+\sum_{i=1}^{n-1}{\left(1-\sum_{t=1}^{n-1}x_{i,t}\right)}^{2}\right\}. \end{array} \end{array}$$

In this new cost function, decision variables $x_{i,t}$ only take value i={1,2,…n} and t={1,2,…n}, and thus we only need effectively ${\left(n-1\right)}^{2}$ decision variables for an *n*-city TSP after fixing the initial city. The reduction in the length of the bit string is especially advantageous because it is ultimately equivalent to reducing the number of qubits for encoding the problem on a quantum circuit. Additionally, it is important to point out that this TSP optimization formulation works for a generally symmetric TSP, not relying on a flat surface, which can be generalized to many real-world applications where non-planar relations are ubiquitous, such as social networks, stock markets, materials science, and so forth. Asymmetric TSP ($\omega_{ij}\neq \omega_{ji}$) can also be formulated similarly in principle but is not considered within the scope of this work.

There are many other ways to formulate *n*-city TSP as an optimization problem [31,32], usually requiring more than $n^{2}$ variables. Recent work [33,34,35] explores unique features of the TSP as an optimization problem and leads to even fewer qubits and computational resources. Within the $n^{2}$-variable formulation, an alternative approach to formulating the TSP expresses the cost function in terms of the adjacency matrix:(11)$$\begin{array}{c} C_{\mathrm{adj}}\left(x\right)=\sum_{0\leq i,j<n}\omega_{ij}x_{ij}^{\mathrm{adj}}, \end{array}$$
where $x^{\mathrm{adj}}$ is the adjacency/connectivity representation of a permutation. The adjacency matrix representation can be particularly useful in symmetric TSP because time degrees of freedom are automatically factored out. Penalty terms for the cost function can be conveniently included by means of the symmetry about the main diagonal. However, unlike our adopted construction, it is not straightforward to reduce the number of decision variables in Equation (Equation 11), and therefore we leave it for a future study. In the subsequent section, we introduce the quantum approximate optimization algorithm based on the improved TSP optimization formulation according to Equation (Equation 10).

## 3. Quantum Approximate Optimization Algorithm (QAOA)

The quantum approximate optimization algorithm (QAOA) [10,23] is a general quantum heuristic approach for solving optimization problems. In this section, we introduce the QAOA workflow in detail and its application to the TSP formulation introduced in Section 2.2.

### 3.1. QAOA Workflow

The QAOA is deeply connected with the adiabatic quantum computation (AQC) [36], which is based on the adiabatic theorem. In AQC, the whole simulation process can be viewed as a time-dependent Hamiltonian evolution represented by H(t), where:(12)$$\begin{array}{c} H\left(t\right)=\left(1-\frac{t}{T}\right)H_{M}+\frac{t}{T}H_{P}. \end{array}$$

Here, $H_{M}$ represents a known ansatz, and $H_{P}$ is the target Hamiltonian that one aims at to find a ground state. According to the adiabatic theorem, by gradually introducing perturbation, an initial eigenstate of $H\left(t=0\right)=H_{M}$ will evolve into the ground state of $H\left(t=T\right)=H_{P}$. However, in practice, simulating this process can be extremely time-consuming, and accurately estimating a suitable duration poses its own challenges. The fundamental idea behind the QAOA is to approximate this adiabatic process by parameterizing the infinitely-long time evolution into finite time steps, addressing practical considerations. In both the original QAOA [10] and the extended QAOA [23], the hybrid quantum approach consists of three essential parts:**State initialization** with initial state |s〉.**Parameterized unitary ansatz** Up(β→,βγ→), a variational ansatz of *p* layers for the TSP, based on two alternating Hamiltonians, $H_{P}$ and $H_{M}$, using respective parameters $\vec{\beta }$ and βγ→.**Measurement and optimization** of the cost expectation 〈β→,βγ→|C(x)|β→,βγ→〉 for the final state |β→,βγ→〉, where an optimizer on a classical computer is used for the minimization.

Putting the three parts together, we construct the complete QAOA circuit, where the final state after the evolution is:(13)|β→,βγ→〉=Up(β→,βγ→)|s〉=∏i=1pUM(βi)UP(γi)|s〉(14)$$\begin{array}{cc} \; & =U_{M}\left(\beta_{p}\right)U_{P}\left(\gamma_{p}\right)\ldots U_{M}\left(\beta_{1}\right)U_{P}\left(\gamma_{1}\right)|s\rangle , \end{array}$$
where *p* is referred to as the depth (or layer number) of the QAOA. Specifically, the two alternating unitary ansatzes in each layer are:(15)$$\begin{array}{c} U_{P}\left(\gamma_{i}\right)=e^{-i\gamma_{i}H_{P}},\;U_{M}\left(\beta_{i}\right)=e^{-i\beta_{i}H_{M}}, \end{array}$$
where $H_{P}$ is the problem Hamiltonian derived from the cost function, and $H_{M}$ is the mixer Hamiltonian that explores the feasible subspace. In this work, we refer to the QAOA ansatz with *p* layers as *p*-QAOA. Note that βγ→ and $\vec{\beta }$ are parameter vectors of length *p* to be optimized, and there is only one single parameter $\gamma_{i}$ ($\beta_{i}$) for the associated unitary ansatz $U_{P}$ ($U_{M}$) per layer. This means there are only two parameters per layer for the QAOA, independent of the number of qubits (i.e., problem size), which makes the approach highly scalable. These parameters or angles can also be regarded as mimicking the Trotterization time steps in the QAOA to approximate the adiabatic evolution in Equation (Equation 12); nonetheless, the behavior in the finite layer limit can be drastically different.

In the last few years, many variants of the QAOA approach have emerged [37]. One such variant is the multi-angle QAOA (ma-QAOA) [38], which uses a unique angle for each element of the Hamiltonian. This approach could potentially reduce circuit depth required for solving the TSP. Another variant, the digitized-counterdiabatic QAOA (DC-QAOA) [39,40] introduces an additional problem-dependent counterdiabatic driving term in each layer to enhance the convergence rate of the optimization process. Additionally, the adaptive-QAOA (ADAPT-QAOA) [41], inspired by the adaptive VQE, systematically selects the mixer ansatz based on the optimization, potentially improving the simulation outcome. Since these more advanced QAOAs generally require more than two parameters per layer and additional simulation time, we opted not to incorporate them in this initial work; however, we have plans to include these variants in a subsequent study, allowing for a more comprehensive analysis of the QAOA to the TSP.

### 3.2. From Binary Decision Variables to Qubits

To carry out the optimization on quantum computers, an efficient qubit encoding scheme is necessary to map the binary decision variable in the TSP formulation to quantum computers. Here, we use the standard boolean binary variable mapping strategy [42]. For an *n*-city TSP, we simply map:(16)$$\begin{array}{c} x_{i,t}\mapsto \left(I_{\left(i,t\right)}-Z_{\left(i,t\right)}\right)/2, \end{array}$$
where $Z_{\left(i,t\right)}$ is the Pauli-Z matrix (see Appendix A) at qubit location (i,t) on a two-dimensional lattice. To identify the qubit on the lattice with its realistic index in a quantum device, one may use the ideal mapping (ignoring the device connectivity) that takes (i,t)→ni+t for the original TSP formulation in Equation (Equation 6). For the improved TSP formulation according to Equation (Equation 10), since both sets of the i=0 and t=0 qubits are never used, we economically map:(17)$$\begin{array}{c} (i,t)\mapsto (n-1)(i-1)+(t-1), \end{array}$$
such that only a total of ${\left(n-1\right)}^{2}$ qubits is needed, from index 0 to ${\left(n-1\right)}^{2}-1$, for *n* cities. Reducing qubit number is crucial in the practical quantum simulation, and therefore we adopt the mapping strategy in Equation (Equation 17) for the improved TSP formulation throughout this work.

### 3.3. State Initialization

The initial states are one of the key components in the QAOA approach. In the original QAOA [10], the initial states are always set to be ${|+\rangle }^{⊗N}$, where *N* is the total number of qubits. For an *n*-city TSP, with the original $n^{2}=N$ case for simplicity, it means the initial state becomes:(18)$$\begin{array}{cc} |s_{H}^{n}\rangle & =H^{⊗n^{2}}|0\rangle ={|+\rangle }^{⊗n^{2}}=\frac{1}{\sqrt{2^{n^{2}}}}\sum_{x=0}^{2^{n^{2}}-1}|x\rangle . \end{array}$$

In this way, the initial quantum state $|s_{H}^{n}\rangle$ is a superposition of all possible basis states for the problem. While this strategy is easy to implement on a quantum device using Hadamard gates H, the magnitude of each basis state in the initial state shrinks exponentially as the number of cities increases because the dimension of the search space grows as $O(2^{N})$.

Recently, additional initialization strategies of a restricted quantum search space following their corresponding mixing ansatzes have been considered in the QAOA. In particular, the so-called $W_{N}$ states [43] can be especially useful as they represent one-hot encoding on the quantum circuit suitable for binary decision variables. For example, a $W_{3}$ state on three qubits is written as:(19)$$\begin{array}{c} W_{3}=\frac{1}{\sqrt{3}}\left(|100\rangle +|010\rangle +|001\rangle \right), \end{array}$$
where each bit string always sums to one. With the property of the *W* state, we can construct an improved initial state to satisfy the temporal or spatial constraints of the TSP automatically, i.e., Equation (Equation 4) or Equation (Equation 5):(20)$$\begin{array}{cc} |s_{W}^{n}\rangle & ={\left(W_{n}|0\rangle \right)}^{⊗n}={\left(\frac{1}{\sqrt{n}}\sum_{i=0}^{n-1}|2^{i}\rangle \right)}^{⊗n}, \end{array}$$
where the temporal constraint is satisfied by putting together multiple *W* states in parallel (technically, there are many ways to build the $W_{n}|0\rangle$ state; we followed the method in Ref. [44]).

With a sufficiently powerful ansatz, one may also consider a permutation initial state, ignoring all superpositions:(21)|sPn〉=|(0,1,…,n−1)〉∑x=1,
where its construction is simplest, using a few Pauli-X gates. We also considered an equal superposition of all permutation states, representing the minimal Hilbert space containing all the valid solutions; however, we found it to be the most challenging to initialize on the circuit.

These choices of initial states provide dramatically different initial search spaces, with dimensions ranging from $O(2^{n^{2}})$, $O(n^{n})$, to O(1), respectively, along with their set relation $\left\{|s_{P}\rangle \}\subset \{|s_{W}\rangle \}\subset \{|s_{H}\rangle \right\}$. Notably, both the $|s_{H}\rangle$ and $|s_{W}\rangle$ are a superposition of solution states, but $|s_{P}\rangle$ is not. The selection of initial states plays a vital role in the QAOA, as it can reduce the number of potential candidates in the quantum evolution, albeit at the expense of an increased number of quantum gates. Lastly, these initial states will be used together with their respective mixer Hamiltonians of the QAOA, which are introduced in the next section.

### 3.4. Variational Ansatzes

Variational ansatzes are essential for optimizing the quantum state to represent the true solution. The variational ansatz $U_{p}$ introduced in Equation (Equation 13) consists of the following two parts.

#### 3.4.1. Problem Hamiltonian

The problem Hamiltonian is the qubitized cost function encoding the specific TSP instance to be solved in the QAOA approach. Specifically, these problem Hamiltonians are obtained by mapping the cost functions (Equations (Equation 6) and (10)) onto the quantum circuit according to the encoding strategy, Equation (Equation 16):(22)$$\begin{array}{cc} C_{\mathrm{dist}}\left(x\right),\;C_{\mathrm{dist}}^{\prime }\left(x\right)\; & \to \;H_{\mathrm{dist}}, \end{array}$$(23)$$\begin{array}{cc} C_{\mathrm{penality}}\left(x\right),\;C_{\mathrm{penality}}^{\prime }\left(x\right)\; & \to \;H_{\mathrm{penality}}, \end{array}$$
where the obtained operators are a sum of the Pauli-Z and Pauli-ZZ operators, known as the Ising Hamiltonian [45]. Combining them, we obtain $H_{P}$, the problem Hamiltonian of the TSP instance:(24)$$\begin{array}{c} H_{P}=H_{\mathrm{dist}}+\lambda H_{\mathrm{penality}}=\sum_{i}c_{i}Z_{i}+\sum_{ij}c_{ij}Z_{i}Z_{j}. \end{array}$$

As a consequence of qubit encoding, a ground state of $H_{P}$ is guaranteed to be a true solution state that minimizes the respective TSP cost function. The Ising representation of the Hamiltonian is easily translated into a quantum circuit using a sequence of quantum gates.

#### 3.4.2. Mixer Hamiltonian

The mixer Hamiltonian defines how the state space is to be explored and impacts how the quantum state evolves significantly with each iteration. Based on the Trotter product formula, the mixer Hamiltonian must not commute with the problem Hamiltonian, $[H_{M},H_{P}]\neq 0$, to simulate a Trottered optimization like the QAOA. Many mixer Hamiltonians have been proposed [24,46,47] for different problems solved via QAOA. For different mixers, appropriate initial states as the eigenstates of the mixer Hamiltonian must be used in accordance with the adiabatic theorem. In evaluating the numerical performance of QAOA for TSP, we consider three types of mixers: X mixer, XY mixer, and row-swap mixer (RS mixer), with details explained below.

(a) The **X mixer** is the original mixer proposed in the QAOA that works together with a number of problems such as the max-cut problem [10]. It takes $s_{H}$ for its state initialization. In the *n*-city TSP, the X mixer is:(25)$$\begin{array}{cc} H_{M_{X}} & =\sum_{i=0}^{n-1}\sum_{t=0}^{n-1}X_{i,t}. \end{array}$$

The X mixer strategy proves most useful for quantum annealing applications, especially on practical D-Wave systems [8]. It is easy to implement on most quantum backends, only requiring $O(n^{2})$ single-qubit X gates per layer in the QAOA.

(b) The **XY mixer** is another natural candidate for the mixing Hamiltonian, preserving the Hamming distance among the acted qubits [48], which is especially suited to the one-hot encoding realized by the initial states $s_{W}$. Here, we construct the XY mixer for the *n*-city TSP as:(26)$$\begin{array}{cc} H_{M_{\mathrm{XY}}} & =\sum_{i=0}^{n-1}\sum_{t=0}^{n-1}{\mathrm{XY}}_{\left(i,t),(i,t+1\right)}, \end{array}$$
where the XY gate is implemented via the Pauli-XX and Pauli-YY gates on the circuit. The block-wise construction allows for the conservation of probability for each city in the TSP, reinforcing the satisfaction of the temporal constraint, as in Equation (Equation 4). A generic XY gate across any two points (i,t) and (j,s) on the 2D lattice is:(27)$$\begin{array}{cc} {\mathrm{XY}}_{\left(i,t),(j,s\right)} & =X_{i,t}X_{j,s}+Y_{i,t}Y_{j,s}, \end{array}$$
where *X* (*Y*) is the Pauli-X (Pauli-Y) matrix. Here, one should understand Equation (Equation 26) as a cyclic iteration of the XY gate. For example, $X_{n-1,n}\equiv X_{n-1,0}$ in the *n*-city case; other variants, such as non-cyclic and fully-connected XY gates, can also be used. The XY gate is often interchangeably referred to as the swap gate, as they both redistribute the amplitudes between two qubits while preserving the total amplitude of the quantum state. Alternatively, one could use the SWAP gate [49] instead of the XY gate to implement the XY mixer via:(28)$$\begin{array}{c} {\mathrm{SWAP}}_{u=(i,t),v=(i,t+1)}=\frac{1}{2}\left(X_{u}X_{v}+Y_{u}Y_{v}+Z_{u}Z_{v}+I_{u}I_{v}\right), \end{array}$$
where a similar performance is produced. Therefore, we choose to use the simpler XY gate to implement the XY mixer throughout this work. Compared with the X mixer, the XY mixer is more expensive to implement by having $O(n^{2})$ XY gates per layer.

(c) The **row-swap (RS) mixer** has recently been proposed in the QAOA as a means of embedding hard constraints directly into the mixer Hamiltonian [23,24]. Although the RS mixer also uses the XY gate, it simultaneously swaps all non-overlapping rows of qubits (corresponding to different cities) as a whole. The RS mixer can be represented as:(29)$$\begin{array}{cc} H_{M_{\mathrm{RS}}} & =\sum_{i=0}^{n-2}\sum_{j=i+1}^{n-1}\prod_{t=0}^{n-1}{\mathrm{XY}}_{\left(i,t),(j,t\right)}, \end{array}$$
where the first two sums represent all possible swapping between city-*i* and city-*j*, and the last product denotes the simultaneous swap of all corresponding entries in the associated cities. In this way, the RS mixer is capable of exploring the entire space of valid solutions when initialized on any single valid state, i.e., $s_{P}$. However, it should be noted that the RS mixer incurs a significant computational cost during the simulation due to the involvement of many tensor products of the Pauli-XX or Pauli-YY matrices. One can mitigate this expense by relying on a set of creation and annihilation operators constructed from four-qubit gates [24]. Nevertheless, the $H_{M_{\mathrm{RS}}}$ ansatz remains computationally expensive, requiring O[(n−1)(n−2)/2] four-qubit gates per layer with each four-qubit gate itself being expensive to construct.

### 3.5. Measurement and Optimization Protocol

Based on the unitary ansatz and its appropriate initial state, the cost expectation of the QAOA is evaluated by measurements performed on quantum devices and subsequently optimized using gradient-free optimizers such as COBYLA [50,51,52] and SPSA [53,54]. The optimization process continues until convergence or until the maximum iteration threshold is reached. The resulting solution to the TSP is then determined by identifying the most dominant quantum state (or binary decision variable encoded in a bit string). To account for statistical fluctuations in measurements, we run each quantum simulation multiple times (typically 5–10) with different random seeds and report the result with the lowest converged expectation value. Considering that the expectation values are TSP-specific, we use the standard evaluation metric called the approximation ratio (AR) to evaluate the performance by normalizing against the ideal cost in different TSPs. The AR is calculated as:(30)AR=simulationcostidealcost=〈β→,βγ→|C(x)|β→,βγ→〉Cideal≥1,
where a lower AR corresponds to a lower expectation cost, indicating a closer approximation to the exact solution. Classical optimizers play a vital role in the optimization, and their advantages can be further utilized in the QAOA. The expectation values of individual bit strings are cached and retrieved on the classical optimizer to enable fast computation of the final cost expectation during each iteration. The option to use constraint bounds of [0,2π) for the ansatz parameters in the case of COBYLA can also accelerate the convergence, which is the main reason we primarily focused on simulations using the COBYLA optimizer in our study, although a comprehensive analysis with other available optimizers can be explored in future research.

To optimize the QAOA, we employed the **layer-wise learning (LL)** protocol introduced in Ref. [28]. In comparison to complete depth learning (CDL), LL proved to be advantageous in reducing the optimization cost, particularly as the number of qubits and circuit depth increased. It also helps mitigate the likelihood of barren plateaus (BP) [28]. In short, the LL is a two-part optimization protocol, as illustrated in Figure 1.

(A)**Progressive pre-training**: In the first part (Figure 1a), we construct the QAOA ansatz by gradually adding layers. Initially, we train and optimize over the leading few layers (typically two layers). Then, for a *p*-layer QAOA simulation, we freeze the parameters in the first (p−1)-th layers, obtained from previous simulations, and exclusively optimize the parameters in the *p*-th layer. Optimal parameters of the current layer that yield the lowest cost expectation are selected. Note the initial values for the parameters of the *p*-th layer are zero. If no lower cost is found at the *p*-th layer compared with previous costs, we use zeros for the parameters of that layer. In this way, the cost is always non-increasing over the entire simulation. This progressive optimization protocol proves to be efficient and leads to an increasingly optimized solution as the number of layers increases. It also reduces the computational cost in parameter searching for very thick layers. We denote this protocol with the letter A and an integer to indicate the depth being optimized.(B)**Randomized retraining**: In the second part (Figure 1b), we take the pre-trained QAOA ansatz from part (A) and randomly select a larger portion of the parameters to be trained at a time. Typically, we free 50% of the parameters in each iteration of retraining. Although it is more computationally expensive, this retraining is still less costly than the CDL, which allows us to train the QAOA ansatz as a whole. This mitigates the risk of becoming trapped in local minima, which could occur when using the protocol of part (A) exclusively. We use the protocol of part (B) with a number to indicate which iteration of retraining is being conducted.

It should be mentioned that there are other variations to the LL, such as sequential block-wise learning used in Ref. [55], where one block/layer is optimized at a time while fixing all other blocks. Layer-wise learning may also be prone to systematic layer saturations [56] that require special treatments, which we leave for future study. Here, our goal is not to prove the superiority of the LL over classical algorithms or any other quantum variational protocols. Instead, we take the LL as a common optimization method to compare the practical performance of various QAOA mixer ansatzes in solving the TSP, which is the highlight of this work. For the numerical results presented in this work, we always use the LL optimization protocol, as its computational cost and solution accuracy consistently outweigh those of the CDL. In Figure 2, we show an example of layer-wise learning applied to the QAOA with the X mixer, showing the optimization in both one protocol step and the full LL procedure; similar performances are also found for other mixers. It is important to point out that the mean square error of the ARs calculated from different batches is negligible, especially toward the end of the optimizations. By contrast, the uncertainty of the ARs calculated from different TSP graphs is significant. The same observation is found for other measurable quantities as well. Therefore, throughout this paper, we exclusively refer to the uncertainty due to various TSP instances.

## 4. Numerical Results

With both the TSP optimization and QAOA method introduced, we perform numerical quantum simulation on the IBM Quantum QASM simulator using aer.QasmSimulator. The problem and mixer Hamiltonian operators are constructed using the qiskit.opflow library. For the circuit implementations of the three mixers, we use Pauli-Z and Pauli-ZZ gates for the X mixer and the XY mixer, and we use the PauliEvolutionGate library for the XY mixer. We focus on quantum simulations using the layer-wise learning protocol for 3-, 4-, and 5-city TSPs on a sufficiently powerful local Ubuntu machine (Ubuntu 22.04.2 LTS machine using one CPU core with 32.0 GB memory and Intel i9 processor of 3.50 GHz), and we compare their performances in terms of numerical accuracy and resource costs. To obtain converged results, we always use a sufficient number of TSP instances, varying from 7 to 10 graphs, depending on the number of cities, mixer, and simulated noise, each with 5–10 repeated runs of quantum simulation.

### 4.1. Simulation Accuracy

We follow the LL optimization protocol introduced in Section 3.5 and use ${\left(n-1\right)}^{2}$ qubits based on the improved TSP formulation (Equation (Equation 10)) for each quantum simulation with *n* cities. In Figure 3, we present the QAOA simulation results when solving various instances of 4-city and 5-city TSPs using the X, XY, and XY mixers. We temporarily leave out the 3-city TSPs in the figure, since all the mixers performed reasonably close to the ideal case. The XY mixer is able to reach an AR of 1.00 given any TSP graph, while the X mixer reaches a value of 1.28 by comparison. The RS mixer is not discussed because any valid solution would be a true solution in the 3-city case for the RS.

The performance is evaluated with three criteria: (a) approximation ratio (AR), (b) percentage of the true solution, and (c) rank of the true solution. The two-part LL optimization is indicated by letters A and B, followed by the specific depth and iteration numbers, respectively. We use a sufficient number of layers in the QAOA simulation (4 layers for 3-city cases and 6 for 4-/5-city cases) to ensure convergence. The uncertainty bars depicted in Figure 3 represent the standard deviations of the respective results calculated for various TSP graph instances. A comprehensive comparison of all the results can be found in Table 1, which includes the results for the 3-city TSP simulations as well.

(a) **Approximation ratio (AR)**: Expectation cost, or equivalently AR, is the primary observable that is measured during the quantum simulation. It directly influences the classical optimizer’s ability to find the optimal parameters. Figure 3a,b demonstrate that both pre-training and retraining parts of the LL are necessary to optimize the AR for various TSP instances. Among the three types of QAOA mixers, the RS mixer achieves the lowest AR, reaching values as low as 1.01±0.01 (in the 4-city case) and 1.18±0.14 (in the 5-city case). On the other hand, the X mixer performs the poorest, particularly as the problem size increases, partially due to the limitations of the ansatz’s expressibility. It is worth noting that even the heuristic VQE ansatz outperforms the X mixer in the 4-city case, with a lower AR around 2.19±0.37, compared to 2.33±0.83 (see Table 1). Considering the temporal constraints during construction, the XY mixer exhibits intermediate performance, with AR values of around 1.44±0.23 and 1.89±0.66 for 4- and 5-city TSPs, respectively.

Lastly, we can also see that all the mixers are only able to reach sub-optimal solutions in the 5-city TSP case. It would be interesting to fully investigate the expressibility of the QAOA ansatzes in obtaining the optimal solution and distinguish that from the optimization itself. We will leave this for an extensive study in the future that involves more TSP cities.

(b) **True percentage**: The percentage of the true solution is also known as the **overlap** between the quantum state and the expected true solution. While the true percentage is determined only after the simulation, it is desirable to have it as large as possible for the accurate extraction of the optimal solution. In Figure 3c,d, we present the true percentages for the three mixers as the TSP problem size increases. Undeniably, RS is the dominating mixer, reaching around 96.3±3.8% and 41.1±29.5%. However, the large uncertainty suggests a highly unstable pattern in the obtained solution; see Appendix B for an explanation. On the other hand, the X mixer gives the lowest percentages, reflecting a poor performance in accurately identifying the true solution. Lastly, the XY mixer is again holding a middle ground, with true percentages of approximately at 36.6±5.7% and 7.4±0.6%, respectively.

(c) **Rank**: The rank of the true solution specifies how many other states possess a higher probability than the state corresponding to the true solution, which is a crucial indicator of the simulation’s accuracy. Achieving a rank of 1 for the true solution signifies consistent identification of the correct solution, as it means the quantum state with the highest probability is always the true solution’s state (so we want to have a rank as low as possible). The results are presented in Figure 3e,f. In the case of the X mixer, it exhibits a significantly high rank, indicating a low likelihood of picking the correct solution among the top quantum states. On the other hand, the ranks of the XY and RS mixers are comparable, both reaching around rank-1 for 4-city TSPs and around rank-2 for 5-city TSPs. Notably, for the 5-city case, we observe lower ranks of the XY mixer in the early stages compared to the final stages, showcasing the effectiveness of the XY mixer in even shallower QAOA for certain TSP instances.

Based on the observations in AR, true percentage, and rank, several conclusions can be made. First, we can see that X mixers consistently under-perform in all three criteria, compared to the other two mixers. This behavior is expected because the Hadamard initialization produces a uniform superposition of all possible states, i.e., $2^{16}$ states in the 5-city case, without any constraints on the solution. As a result, it becomes challenging for the classical optimizer to filter out the invalid and false solutions based solely on the problem Hamiltonian. In particular, when the problem size increases, the X mixer alone is not suitable for the QAOA simulation of the TSP. Secondly, we observe that the RS mixer stands out as the dominating mixer in terms of AR and true percentage, which makes it a reliable candidate for QAOA. In terms of rank, the performances of the XY and RS mixers are quite similar. The strategies employed by the two mixers are very different: the RS mixer relies heavily on the expressibility of the mixer itself, while the XY mixer combines the initialization and the mixing Hamiltonian to achieve its results. By utilizing a single-bit string as the initial state, RS may potentially overlook the benefits of having superposition states in a quantum simulation. In a sense, the XY mixer takes a more balanced approach, whereas the RS mixer takes a more assertive approach; this distinction between the two mixers can have implications for the resource cost, which is discussed in the following section.

### 4.2. Resource Evaluations

Besides numerical accuracy, resource cost estimation is another crucial factor to consider in quantum simulation, as any computational resource is always finite. On the quantum computer and simulator, many factors will contribute to the performance of the simulation, including attributes of the transpiled quantum circuits, such as the number of qubits, the number of single-qubit (double-qubit) gates, and the quantum circuit depth. In a practical calculation, properties of the quantum device, such as qubit connectivity, coherent error, and incoherent noise, will also come into play. For this section, we focus on the quantum circuits of the three QAOA mixers and compare their resource costs on ideal devices; practical calculation is discussed in the subsequent section using noisy simulation.

In Table 2, we compare the properties of the quantum circuits of the three mixers after transpilation for both finite and generic TSP cases. As expected, the complexity of the circuit, measured in terms of quantum gates and circuit depth, generally increases with the number of cities, resulting in a longer simulation time. Notably, the RS mixer incurs a significantly higher resource cost compared to the X and XY mixers, as reflected in the simulation time in practice. As discussed earlier, this increased cost is primarily attributed to the utilization of four-qubit gates in the RS mixer, leading to a quadratic scaling, i.e., $O(n^{4})$, of single-qubit and double-qubit gates. The abundance of double-qubit gates is anticipated to pose serious challenges in executing the simulation on a real quantum device or when employing a noise model [58]. Interestingly, despite requiring fewer resources, the X mixer actually takes a longer time to run in practice compared to the XY mixer, particularly as the number of qubits increases. This observation is likely due to the computational burden of the optimizer when evaluating the expected cost for a dense superposition of bit strings. On the other hand, the XY mixer requires relatively low computational resources, scaling linearly with the circuit depth and quadratically with the number of quantum gates, which is a more economical choice for running QAOA simulations. Considering both optimization accuracy and computational cost, the XY mixer emerges as a more balanced choice for the QAOA. Nonetheless, a resource cost of $O(n^{2})$ gates and qubits for the XY mixer is still quite expensive as *n* increases. Notably, building the XY mixer at the pulse level [59] has the potential to further enhance its numerical performance. Lastly, it should be acknowledged that the resource costs of all mixers would be even higher when simulating on current NISQ or future fault-tolerant quantum computers. In the interest of addressing this aspect, we present noise-model simulations in the subsequent section.

### 4.3. Robustness against Noise

Estimating the performance of the QAOA simulation in the presence of noise is crucial to implementations on NISQ and fault-tolerant devices in the future. In this section, we employ the NoiseModel class from Qiskit to study the sensitivity of the simulation on different noise levels. In particular, we focus on noisy QAOA simulations with XY and RS mixers for the same set of 4-city TSP problems. In Figure 4, we compare the performance of various noise simulations in terms of AR, true percentage, and rank. We consider noise models with different degrees of single-qubit errors: 0.005%, 0.01%, 0.05%, and 0.1%. Besides single-qubit errors, we set the double-qubit errors to be 10 times their respective single-qubit errors, which is a reasonable approximation for realistic two-qubit gates such as the CX gate. For the current study, we have omitted other potential errors for simplicity, such as the qubit connectivity and thermal relaxation time, which can also be implemented with the noise model.

From the results presented in Figure 4, it is evident that the gate errors in the noise model directly impact the quality of the simulation. As expected, QAOA simulations with larger errors perform poorly compared to smaller ones. Interestingly, there seems to be a noise threshold in the simulation results: noisy simulations with error less than or equal to 0.01% exhibit qualitatively different behavior compared to those with higher errors, as shown in Figure 4a–c. Additional details of the noisy simulation are provided in Table 3, where we can clearly see that the LL protocol fails to optimize the QAOA simulation at 0.1% and 0.05% noise levels. Comparing the two ansatzes, the XY mixer outperforms the RS mixer in all indicators for all noisy simulations. Surprisingly, the XY mixer achieves performance similar to the ideal simulation with errors less than or equal to 0.01%, indicating its potential resilience against simulation noise. Our result suggests that the XY mixer is a more suitable choice among the three mixers when considering noise effects in the QAOA simulation.

### 4.4. Problem Dependence

It is important to investigate the problem dependence of the QAOA simulation of the TSP in preparation for the full-fledged quantum simulation. Here, we study several TSP problem dependencies, such as the topology of the TSP graphs and the penalty weight. The topology of the TSP graph could potentially have a significant impact on the performance of the quantum simulation algorithm. One characteristic we consider is the “skewness” of the TSP graphs, which represents the level of asymmetry. To measure the skewness, we analyze the distribution of the edge weights $\omega_{ij}$ in the graph using Fisher–Pearson’s moment coefficient [60,61]. Specifically, we calculate the skewness parameter *g* by:(31)$$\begin{array}{c} g=\frac{m_{3}}{m_{2}^{3/2}},\;m_{k}=\sum_{0\leq i<j<n}\frac{{\left(\omega_{ij}-\bar{\omega }\right)}^{k}}{\left|\omega \right|}, \end{array}$$
where $\bar{\omega }$ is the mean of the edge weights, and |ω| is total number of edges in the graph. Here, $m_{3}$ is the third moment of the edges, and $m_{2}$ is the variance, the square of the standard deviation. Intuitively, the skewness can also be computed as the average value of the cubed z-scores. For instance, a skewness value of 0 indicates a symmetric/normal distribution of the edge, and skewness values of greater than 1 or less than −1 typically indicate highly-skewed distributions. Negative (positive) skewness indicates a left-skewed/right-leaning (right-skewed/left-leaning) distribution.

In Figure 5, we present the quantum simulation using X, XY, and RS mixers on various 4-city TSP graphs with varying skewnesses. Here, we focus on the approximation ratio in the final step of layer-wise learning to assess the dependence on the TSP graph’s skewness. Taking every bit string solution into account, AR represents the overall effectiveness of the simulation, which is suitable to analyze the skewness. We observe that the simulation tends to perform less effectively with right-skewed edge distributions, possibly due to the presence of more low-weight edges in positively-skewed graphs. Further investigations that include sampling uncertainties are necessary to fully study the consequences of varying graph topology for TSPs with more cities.

Additionally, the penalty weight λ in the TSP cost equation (Equation (10)) is essential to examine, for it directly controls the gaps between valid and invalid solutions. By a similar analysis for the skewness, we find that the simulation performs optimally when λ is in the range of $[1.0E_{G,\max}$, $4.5E_{G,\max}]$, where $E_{G,\max}$ represents the maximum TSP edge weight. This analysis further supports the choice of the penalty weight used in this study.

## 5. Summary and Discussions

In this paper, we solved the symmetric TSP (traveling salesman problem) as an optimization problem by using three distinct ansatzes to the QAOA (quantum approximate optimization algorithm) approach. By adopting a layered learning optimization protocol, we performed numerical quantum simulations on gate-based quantum simulators for various 3-, 4-, and 5-city TSPs. In particular, we presented and compared the performance of the three types of mixer ansatzes for the QAOA: the X mixer, the XY mixer, and the RS mixer. For the few-city TSPs studied in this work, we demonstrated that a well-balanced quantum simulation, such as using the XY mixer, is potentially more suitable in terms of both numerical accuracy and computational cost. These findings are further validated through the noise model simulations. Additionally, we highlighted other factors that may play a role in the quantum simulation, such as the TSP graph skewness and cost function penalty.

Our research is a significant step towards finding a successful strategy for the TSP optimization problem using the gate-based QAOA approach, which is particularly interesting for the current NISQ paradigm. The QAOA simulation complements traditional quantum annealing methods in the infinite time region, where efficient qubit reduction techniques, improved optimization protocols, and resource-efficient mixer ansatzes investigated in this work are expected to be valuable for realistic quantum device simulations. Moving forward, we plan to extend our investigations to larger-city TSPs, employing deeper QAOA circuits on noisy quantum backends. By utilizing an adaptive shot-frugal optimizer [62] and implementing digitized-counterdiabatic quantum approximate optimization methods [39,40], we aim to further enhance the accuracy and efficiency of our TSP simulations.

## Figures and Tables

**Figure 1 entropy-25-01238-f001:**
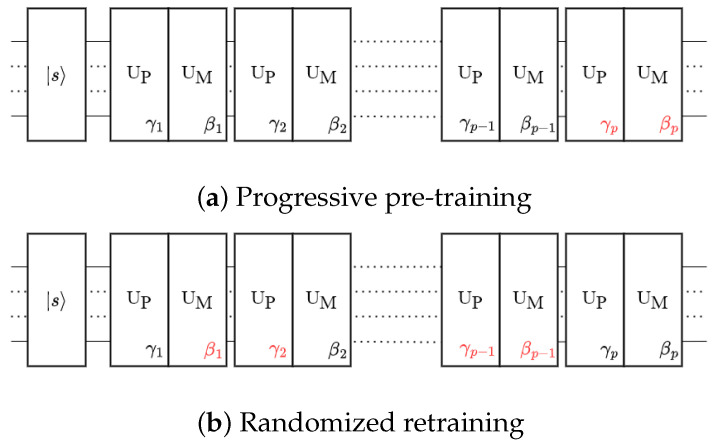
Two-part layer-wise learning protocol of the QAOA. Horizontal lines represent the qubits; rectangular boxes are the unitary operators. Fixed parameters are in black; free parameters are in red.

**Figure 2 entropy-25-01238-f002:**
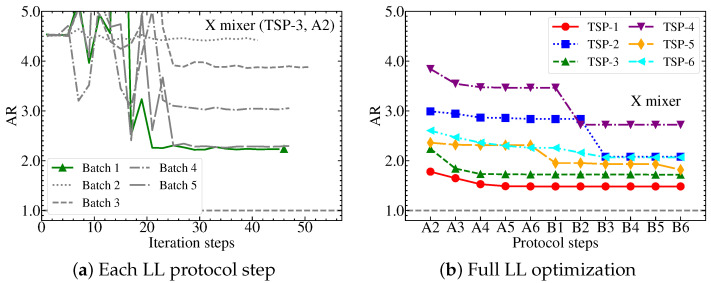
Example of the layer-wise learning protocol applied to the QAOA X mixer simulation. The left panel (**a**) shows the simulation for a selected LL protocol step A2 of a specific TSP instance, TSP-3. The right panel (**b**) shows the overall LL optimization for 6 different TSP instances. Here, a TSP instance is a random TSP graph with 3 nodes and a maximum edge weight of 20.

**Figure 3 entropy-25-01238-f003:**
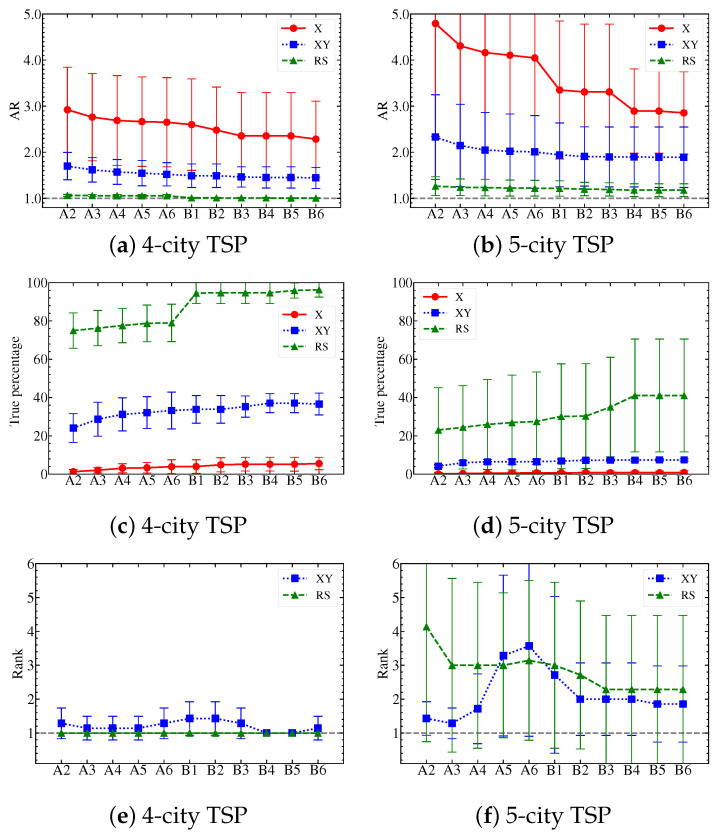
Performance comparison of the 3 QAOA mixers for samples of the 4-city TSP (**left column**) and 5-city TSP (**right column**). In both cases, we compare the AR in panels (**a**,**b**), the percentage of the true solution in panels (**c**,**d**), and the rank of the true solution in panels (**e**,**f**). The uncertainty bars are standard deviations obtained from simulations of different TSPs. Notably, we leave out the ranks for the X mixers due to their significantly higher values, which further complicates the presentation.

**Figure 4 entropy-25-01238-f004:**
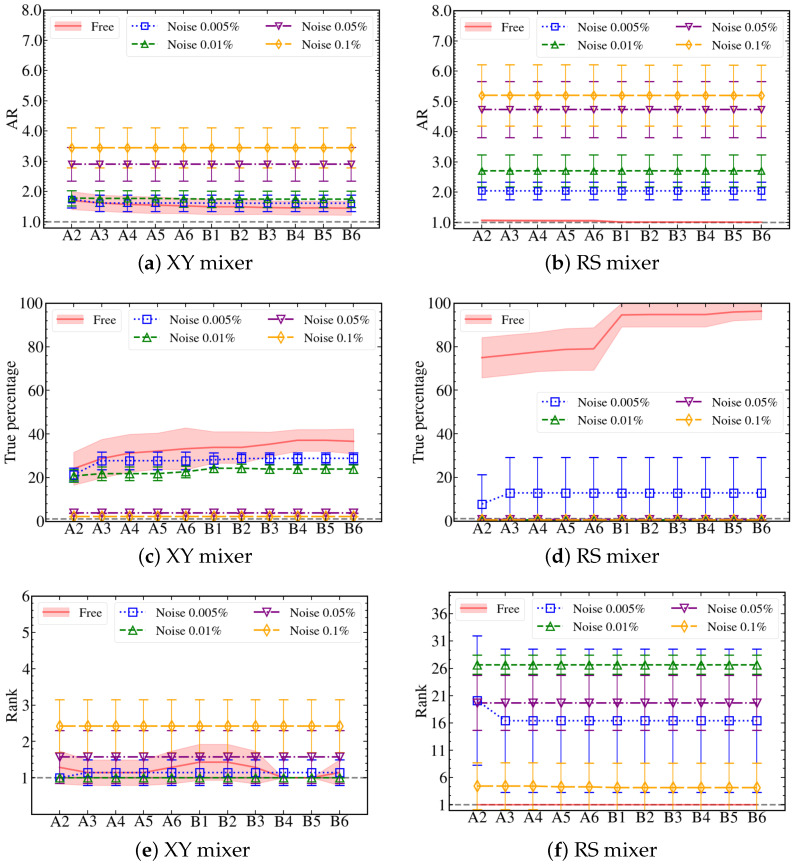
Noisy QAOA simulation results of the XY and RS mixers compared with the noise-free simulation of the 4-city TSP graph. In the legend, we show the single-qubit error used for each noisy simulation. The uncertainty bars/bands are standard deviations obtained from simulations of different TSPs. The same scale is used for XY and RS, except for the plot of their ranks.

**Figure 5 entropy-25-01238-f005:**
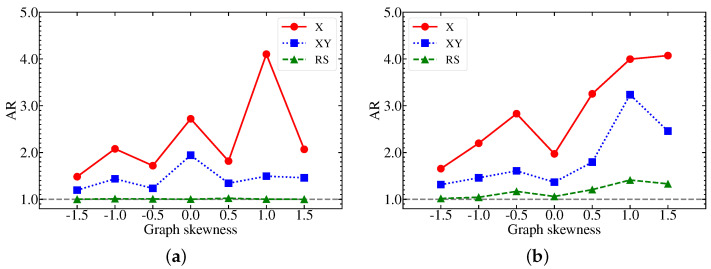
Results of the approximation ratio (AR) of the three QAOA mixers for the 4-city TSP (**panel a**) and the 5-city TSP (**panel b**) of distinct graph skewnesses.

**Table 1 entropy-25-01238-t001:** Comprehensive comparison of the numerical accuracy for the QAOA mixers and heuristic ansatzes used to solve the TSP. The standard deviations of the quantities obtained from variation in TSP graphs are provided in the parentheses. Problem-specific VQE to the TSP such as in Ref. [57] may produce a significantly better result. Notably, the RS mixer is excluded for the 3-city TSP because any starting bit string is a true solution for the RS, which makes it trivial to simulate.

		After Pre-Training			After Retraining		
**# City**	**Mixers**	**AR**	**True %**	**Rank**	**AR**	**True %**	**Rank**
3	VQE	1.02 (0.02)	87.6 (28.6)	1.1 (0.3)	1.01 (0.02)	90.9 (20.2)	1.1 (0.4)
	X	1.34 (0.14)	38.5 (24.8)	2.3 (2.2)	1.28 (0.10)	44.2 (28.2)	1.9 (1.4)
	XY	1.00 (0)	100.0 (0)	1.0 (0)	1.00 (0)	100.0 (0)	1.0 (0)
4	VQE	2.23 (0.29)	12.8 (13.3)	35.6 (44.1)	2.19 (0.37)	11.8 (13.9)	39.1 (42.2)
	X	2.65 (0.97)	3.9 (3.6)	67.7 (140.8)	2.33 (0.83)	5.5 (3.2)	4.7 (5.3)
	XY	1.52 (0.25)	33.2 (9.6)	1.3 (0.5)	1.44 (0.23)	36.6 (5.7)	1.1 (0.4)
	RS	1.05 (0.03)	79.0 (9.8)	1.0 (0)	1.01 (0.01)	96.3 (3.8)	1.0 (0)
5	X	4.05 (2.10)	0.7 (0.8)	1814.1 (4348.8)	2.85 (0.90)	0.9 (1.0)	94.3 (105.0)
	XY	2.01 (0.79)	6.5 (1.3)	3.6 (2.7)	1.89 (0.66)	7.4 (0.6)	1.9 (1.1)
	RS	1.22 (0.17)	27.6 (25.8)	3.1 (2.4)	1.18 (0.14)	41.1 (29.5)	2.3 (2.2)

**Table 2 entropy-25-01238-t002:** Quantum resource estimation per QAOA layer of various mixers considered in the 3-, 4-, and 5-city TSPs. The circuit depth, the count of single-qubit gates, and the count of the double-qubit gates are evaluated after transpilation (light transpilation, no approximation with qiskit.compiler.transpile) to the standard basis gate sets {CX, I, RZ, SX, X} used by the IBM Quantum. Exact numbers for the circuit depths and quantum gates are obtained whenever available; otherwise, asymptotic scalings are provided. Estimating the circuit depth exactly is difficult for the RS mixer. In comparison, it appears to increase linearly with a large slope of 1181 for small city numbers.

# City	# Qubits	Mixers	Circuit Depth	Single-Qubit Gates	Double-Qubit Gates
3	4	X	5	20	0
		XY	26	64	16
4	9	X	5	45	0
		XY	37	144	36
		RS	668	477	432
5	16	X	5	80	0
		XY	48	256	64
		RS	1553	1808	1728
*n*	${\left(n-1\right)}^{2}$	X	5	$5n^{2}$	0
		XY	26+11(n−3)	$O(n^{2})$	$4n^{2}$
		RS	${O(n)}^{1}$	$O(n^{2}{\left(n-1\right)}^{2})$	$O(n^{2}{\left(n-1\right)}^{2})$

**Table 3 entropy-25-01238-t003:** Details of noisy simulation for the 4-city TSP case. Noise percentage refers to single-qubit errors used in the noise model simulation. The standard deviation of the quantities obtained from variation in TSP graphs is provided in parentheses.

		XY Mixer			RS Mixer		
**Noise %**	**Protocol**	**AR**	**True %**	**Rank**	**AR**	**True %**	**Rank**
0.1	A2	3.44 (0.67)	2.09 (0.03)	2.43 (0.73)	5.20 (1.02)	0.4 (0.1)	4.4 (4.3)
	A6	3.44 (0.67)	2.09 (0.03)	2.43 (0.73)	5.20 (1.02)	0.4 (0.1)	4.3 (4.4)
	B6	3.44 (0.67)	2.09 (0.03)	2.43 (0.73)	5.19 (1.00)	0.4 (0.1)	4.1 (4.5)
0.05	A2	2.90 (0.56)	3.7 (0.1)	1.6 (0.7)	4.73 (0.92)	0.6 (0.1)	19.7 (5.1)
	A6	2.90 (0.56)	3.7 (0.1)	1.6 (0.7)	4.73 (0.92)	0.6 (0.1)	19.7 (5.1)
	B6	2.90 (0.56)	3.7 (0.1)	1.6 (0.7)	4.73 (0.92)	0.6 (0.1)	19.7 (5.1)
0.01	A2	1.78 (0.25)	20.8 (2.9)	1.0 (0)	2.70 (0.53)	0.6 (0)	26.7 (1.8)
	A6	1.76 (0.27)	22.7 (2.7)	1.0 (0)	2.70 (0.53)	0.6 (0)	26.7 (1.8)
	B6	1.74 (0.27)	23.9 (2.0)	1.0 (0)	2.70 (0.53)	0.6 (0)	26.7 (1.8)
0.005	A2	1.74 (0.29)	21.3 (3.1)	1.0 (0)	2.04 (0.29)	7.7 (13.5)	20.1 (11.8)
	A6	1.62 (0.27)	27.8 (4.0)	1.1 (0.4)	2.04 (0.29)	12.9 (16.3)	16.4 (13.1)
	B6	1.61 (0.28)	28.8 (2.5)	1.1 (0.4)	2.04 (0.29)	12.9 (16.3)	16.4 (13.1)

## Data Availability

Not applicable.

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
