# Peer review of "Comparative Study of Variations in Quantum Approximate Optimization Algorithms for the Traveling Salesman Problem"

_entropy, 2023, doi:10.3390/e25081238_

Round 1

Reviewer 1 Report

This paper presents a quantum approach to the Traveling Salesman Problem (TSP) using the quantum approximate optimization algorithm (QAOA). The paper explores different qubit encoding and variational ansatz strategies to reduce the problem complexity and improve the simulation accuracy. The paper evaluates the proposed approach on various TSP instances and compares it with classical methods. The paper claims that the proposed approach can achieve better results for some TSP instances and can be extended to other combinatorial optimization problems.

The authors present an intriguing approach, blending reinforcement learning with variational algorithms. However, a more thorough, standard learning procedure is notably absent, with the authors instead proposing a variational process that requires further justification. Several key concerns must be resolved before the work can be seriously reconsidered:

  1. The work lacks a robust comparative analysis, demonstrating that the proposed Q-algorithm consistently outperforms traditional ones. For a solid confirmation of superiority, comprehensive testing against established, classical algorithms is recommended.
  2. The chosen optimization process is oriented towards achieving an 'ideal cost' rather than minimizing the Mean Square Error (MSE). This raises questions about the theoretical limitations of the approach, particularly in scenarios where the 'ideal cost' is significantly high. Even if the Q-circuit can yield results in such situations, can these results truly be considered valid solutions to the Traveling Salesman Problem (TSP)?
  3. Based on the AR vs Iterations data, it appears that the optimization of XY and RS gates is suboptimal. Could the authors elucidate on the extent to which these quantum gates contribute to resolving the TSP during the variational process? A quantitative analysis would substantially enhance this aspect of the study.
  4. The concept of the "True Percentage" lacks clarity. Is it intended to represent the error between the best and generated solution? If it instead denotes the discrepancy between the 'ideal' solution and the generated one, this could significantly undermine the validity of the conclusions drawn. Thus, a more precise definition and justification for the "True Percentage" is warranted.

Reviewer 2 Report

In this study, the authors addressed the symmetric Traveling Salesman Problem (TSP) using the Quantum Approximate Optimization Algorithm (QAOA) with three distinct ansatzes: the X mixer, the XY mixer, and the RS mixer. The research was conducted via numerical quantum simulations on gate-based quantum simulators for 3-, 4-, and 5-city TSPs. The QAOA works by applying a sequence of unitary operators (which are functions preserving the length in complex Hilbert space) to a starting state, typically the equal superposition of all computational basis states. The choice of these operators is guided by the problem one is trying to solve. This creates an ansatz (trial) state, which is a superposition of all possible solutions to the problem. One key advantage of QAOA is that it requires relatively short-depth circuits, making it more amenable to near-term quantum devices, which are often noisy and have limited coherence times. The study found that a balanced quantum simulation, such as the one using the XY mixer, was potentially more suitable in terms of numerical accuracy and computational cost. The paper emphasized other influencing factors in the quantum simulation, including the TSP graph skewness and cost function penalty. This research represents a significant stride toward devising a successful strategy for TSP optimization through the gate-based QAOA approach. As a future plan, the researchers aim to extend their work to larger-city TSPs, using deeper QAOA circuits on noisy quantum backends, while also implementing digitized-counterdiabatic quantum approximate optimization methods.

Other comments:

I think that the choice of 3-city TSP instances should be justified with a short paragraph. Such instances either have a single Hamiltonian cycle or none at all.

In general, TSP instance are not necessarily complete graphs and not necessarily Euclidean. Finding a single Hamiltonian cycle is therefore NP-Hard. Any approximation algorithm, whether quantum or classical, that approximates TSP must therefore solve the NP-Hard problem of finding a Hamiltonian cycle.

I think that this paper will be of interest to the people working in the area. The English, and writing style, are very good and I did not find any typos in the script.

I look forward to the results when this work is extended to higher-cardinality TSP instances.

Reviewer 3 Report

This manuscript presents a solid comparative study on solving TSP using QAOA. The study includes detailed study on accuracy, complexity analysis and robustness to noise. I think this manuscript presents a solid study and is almost good enough to be accepted in current format.

Just a minor comment, in the results part, some mixers are missing for various comparison, for example, Figure 3 missed mixer x in (e) and (f), Table 2 missed RS for the first row. It would be better to have all three mixers presented in each tables/figures.

Round 2

Reviewer 1 Report

No more comments.